# A Machine Learning Approach to Predicting Academic Performance in Pennsylvania's Schools

**Shan Chen [1] and Yuanzhao Ding [2,***

1 Department of Applied Social Sciences, The Hong Kong Polytechnic University, 11 Yuk Choi Rd., Hung Hom, Hong Kong, China
2 School of Geography and the Environment, University of Oxford, South Parks Road, Oxford OX1 3QY, UK
* Correspondence: armstrongding85@gmail.com

**Abstract:** Academic performance prediction is an indispensable task for policymakers. Academic performance is frequently examined using classical statistical software, which can be used to detect logical connections between socioeconomic status and academic performance. These connections, whose accuracy depends on researchers' experience, determine prediction accuracy. To eliminate the effects of logical relationships on such accuracy, this research used 'black box' machine learning models extended with education and socioeconomic data on Pennsylvania to predict academic performance in the state. The decision tree, random forest, logistic regression, support vector machine, and neural network achieved testing accuracies of 48%, 54%, 50%, 51%, and 60%, respectively. The neural network model can be used by policymakers to forecast academic performance, which in turn can aid in the formulation of various policies, such as those regarding funding and teacher selection. Finally, this study demonstrated the feasibility of machine learning as an auxiliary educational decision-making tool for use in the future.

**Keywords:** machine learning; neural network; socioeconomic status; population; crime rate; academic performance

## 1. Introduction

For a considerable duration, educational systems have widely utilized standardized examinations as a large-scale means of effectively sorting students. When it comes to evaluation efficiency, standardized test scores are overwhelmingly superior in identifying talent over other qualities that schools ought to place greater emphasis on, such as moral character, life adaptability, non-cognitive skills, and social responsibility (Ebel and Frisbie 1972). These preferences are rooted in the strengths of standardized tests, which are a product of historical and social conventions. There are considerable and obvious advantages to employing paper-and-pencil examinations that feature a series of archetypal questions, including practicality, reliability, good content validity, convenience, accessibility, and openness.

Despite the usefulness of traditional tests in assessing students' knowledge and skills, there are several other factors that can impact academic performance, often overlooked. One significant factor identified in predictive studies is socioeconomic status (SES), which plays a vital role in widening the academic performance gap between students in rural and urban institutions (Ramos et al. 2012). In some European countries, high SES often correlates with above-average exam scores, highlighting the significant impact of SES on educational performance (Jana et al. 2006; Willms et al. 2006). Conversely, in eastern Europe, low SES and students from rural schools can negatively affect academic performance (Kryst et al. 2015). While some studies found few differences between rural and urban schools (Miller et al. 2019) or no significant distinctions among students from different school settings (Fan and Chen 1998). Many researchers and educators continue to explore the effects of SES

on academic performance using correlation and regression analysis. As such, this study aims to employ machine learning (ML) models, a novel approach in predictive studies, to investigate the impact of SES on student academic performance. Numerous studies have demonstrated the considerable accuracy of ML prediction compared to a classic statistical method such as correlation and linear regression (Table 1) (Chang et al. 2020; Paulick et al. 2013). As an artificial intelligence approach, ML has had a far-reaching influence on handling the vast amounts of facts and numerical data generated by computers through simulations of the human brain. For instance, an ML algorithm is superior in analyzing considerable internet data than regular models, since it enables relatively rapid prediction with high accuracy and large datasets (Fedushko and Ustyianovych 2019; Shakhovska et al. 2017; Zhou et al. 2017). Applying ML algorithms also enables researchers and teachers to recognize the key factors that strongly influence student performance and find more effective ways to improve teaching quality (Buenaño-Fernández et al. 2019; Hussain et al. 2019; Kemper et al. 2020). The problem is that previous studies were a small-scale, incomprehensive and restricted data pool to address certain groups under limited conditions. This scope cannot ensure overall effective outcomes of ML prediction, and a large representative sample has yet to be used to further verify the precision of ML results.

**Table 1.** Comparison of a classical statistical method (e.g., correlation and linear regression) and ML.

|  | Classical Statistical Method | ML Method | Reference |
|---|---|---|---|
| **Rationale** | Necessary for understanding the relationship between academic performance and relevant factors (e.g., crime rate and population density) | Prediction of academic performance by ML algorithms | (Bujang et al. 2021; Chang et al. 2020; Lykourentzou et al. 2009; Mduma et al. 2019; Papernot et al. 2017; Paulick et al. 2013; Şara et al. 2015) |
| **Methods** | The use of programs such as Mplus to identify relationships between academic performance and relevant factors; calculation based on the relationships | Prediction via 'black box' models without consideration for relationships | (Bujang et al. 2021; Chang et al. 2020; Lykourentzou et al. 2009; Mduma et al. 2019; Papernot et al. 2017; Paulick et al. 2013; Şara et al. 2015) |
| **Accuracy** | Existing relationships and assumptions | Quality and quantity of data | (Al-Jarrah et al. 2015; Ciolacu et al. 2017; Sekeroglu et al. 2019) |
| **Advantages** | Matured methods with clear processes | Rapid and convenient prediction for reasonable results | (Ciolacu et al. 2017; Sekeroglu et al. 2019) |
| **Limitations** | Sample selection bias | The 'black swan' effect | (Batrouni et al. 2018; Lorey et al. 2011) |

In ML prediction, different variables may strongly affect student performance. In this respect, Musso et al. administered a questionnaire on digital tools, health, social support, demographic items, cognitive attributes, and learning and coping strategies and used a neural network algorithm to predict student performance (Musso et al. 2020). Qazdar et al. incorporated several variables, such as gender, test score, and performance, into the forecasting of students' test results (Qazdar et al. 2019). Yousafzai et al. use a digital management system (which reflects student information and academic progress) to their advantage in predicting test scores (Yousafzai et al. 2020). The decision tree and KNN models used by the authors achieved an accuracy of 85%. Alyahyan and Düştegör explored the factors that contribute to successful performance in academics (e.g., sociodemographic, psychological and academic factors, and cognitive qualities) (Alyahyan and Düştegör 2020). Boxer et al. discovered a negative relationship between crime and student performance in language and math, with impoverished students engaged in more delinquencies and criminal events (Boxer et al. 2020). Although there are many factors play a role in the prediction of student performance, sociodemographics, and crime rates have an important influence on academic performance in schools.

Above all, the present research focused on the effects of socioeconomic status (SES) and crime rates on school performance. The study chose Pennsylvania as the basis for our study and an ML model to predict academic performance in the state. Using population, crime, and school data, this study trained five ML models: a decision tree, random forest, logistic regression, support vector machine, and neural network (Figure 1). Among these models, the neural network could predict overall academic performance in schools precisely, despite the significant deviations among individual students, such as abnormal performance in examinations. This study also demonstrated the capability of the neural network to identify which factors (e.g., crime rate) are the most important in affecting academic performance. In summary, this work pointed to the feasibility of the ML model as an auxiliary tool for decision making in the future.

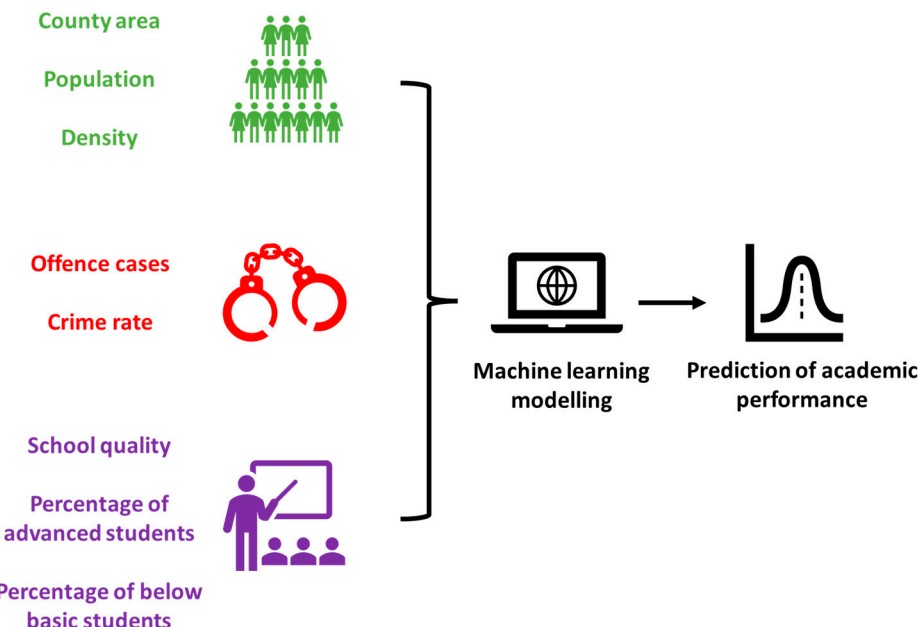

**Figure 1.** Framework of this study shows how ML used big data to predict academic performance.

## 2. Materials and Methods

### 2.1. Data Collection

Pennsylvania is selected as the model area because (1) Pennsylvania provides a full set of online education data; (2) Pennsylvania is a representative state that includes megacities and rural areas; and (3) Pennsylvania is a state with a wide range of educational resources. The educational data are downloaded from Pennsylvania School Performance Profile (https://paschoolperformance.org/, accessed on 1 December 2022), including "Grade", "School level", "Sample size", "Subject", "Percent of advanced student", and "Percent of below-basic student". The county data are obtained from the United States Census Bureau (https://www.census.gov/, accessed on 1 December 2022) and World Population Review (https://worldpopulationreview.com/us-counties/states/pa, accessed on 1 December 2022), such as "County area", "County population" and "County density". The crime data are taken from Pennsylvania uniform crime reporting system (UCR) records (https://www.attorneygeneral.gov/, accessed on 1 December 2022), such as "Total offence cases", and "Crime rate". The rural-urban definitions are referred to Center for Rural Pennsylvania (https://www.rural.pa.gov/data/rural-urban-definitions, accessed on 1 Dec 2022).

Based on our classifications presented in Table 2, we performed statistical analysis and ML calculations.

**Table 2.** Classification method for data treatment.

| County Area (km²) | Approximate Number | County Population (People) | Approximate Number | Crime Rate (per 1000 People) | Approximate Number |
|---|---|---|---|---|---|
| ≤1000 | 500 | ≤50,000 | 25,000 | (3, 6] | 5 |
| (1000, 2000] | 1500 | (50,000, 200,000] | 100,000 | (6, 10] | 8 |
| (2000, 3000] | 2500 | (200,000, 1,000,000] | 500,000 | (10, 16] | 13 |
| (3000, 4040] | 3500 | (1,000,000, 1,584,064] | 1,500,000 | (16, 30] | 29 |

| Population Density (people/km²) | Approximate Number | Total Offenses Cases | Approximate Number | Percentage of Advanced/Below-Basic Students | Approximate Number |
|---|---|---|---|---|---|
| ≤100 | 50 | ≤10,000 | 5,000 | 0% | 0% |
| (100, 500] | 300 | (10,000, 50,000] | 25,000 | (0%, 10%] | 5% |
| (500, 1300] | 900 | (50,000, 200,000] | 100,000 | (10%, 20%] | 15% |
| (1300, 4564] | 3000 | (200,000, 859,411] | 500,000 | (20%, 40%] | 30% |
|  |  |  |  | (40%, 60%] | 50% |
|  |  |  |  | (60%, 100%] | 80% |

| Subject | Assigned Number | School Level | Assigned Number | Rural/Urban | Assigned Number |
|---|---|---|---|---|---|
| English language | 1 | Historically under performance | 1 | Rural | 1 |
| Math | 2 | All group | 2 | Urban | 2 |
| Science | 3 |  |  |  |  |

### 2.2. ML Models

More than 33,000 educational records were input into the ML models. Unless otherwise stated, the split ratio was 76–24% for train-test sets. After the model was trained, this study applied this model to make predictions on the data in unknown areas (see Figure 1). The authors trained the ML via Anaconda 3 and Jupyter 6.3.0 platform. The python coding library was based on scikit-learn (sklearn), keras, pandas, and matplotlib. Five ML methods were compared: decision tree (Somvanshi et al. 2016), random forest (Liu et al. 2012), logistic regression (Rymarczyk et al. 2019), support vector machine (Somvanshi et al. 2016), and neural network (Jung and Kim 2016; Qi et al. 2019). ML methods were based on previous studies with default settings in the sklearn module (Chen and Ding 2022; Pedregosa et al. 2011).

The neural network consisted of 100 hidden layers, each with 100 nodes. The maximum number of iterations was 50. The activation function was the rectified linear unit (relu). The solver for the neural network algorithm was adam optimization ("adam"). The python coding for training, testing, and prediction was also attached to the supplemental information (Pomerat et al. 2019). ML coding was attached to supplemental material Table S1. The tuning process followed the random search method (Table S2) and tuning result was attached to Table S8.

Four Pennsylvania heatmaps were drawn by RStudio: county population, total offense cases, percentage of advanced students (real situation), and percentage of advanced students (neural network prediction). The RStudio coding was based on packages of tidyverse, readr, and maps. The color bar followed terrain.colors, and heat.colors. R coding for the heatmap was attached in the Supplemental Material Tables S3–S7.

For this study, Dell Inspiron 15 TGL 3000 with Intel CoreTM i7-1165G7 CPU and 16 GHz 3200 MHz memory was used unless otherwise stated. The total calculation time to run the coding was around 1–2 h for each round. The computing power of the computer significantly affected performance. High-performance computing was required when running the coding (Correa-Baena et al. 2018).

## 3. Results

### 3.1. Educational Data Analysis

The correlation heatmap was shown in Figure 2. From the correlation heatmap, we found that the population density, total offense cases, and crime rates had a strong positive correlation with each other. A higher population in the county normally implied higher density (+0.85), more total offense cases (+0.90), and a higher crime rate (+0.79).

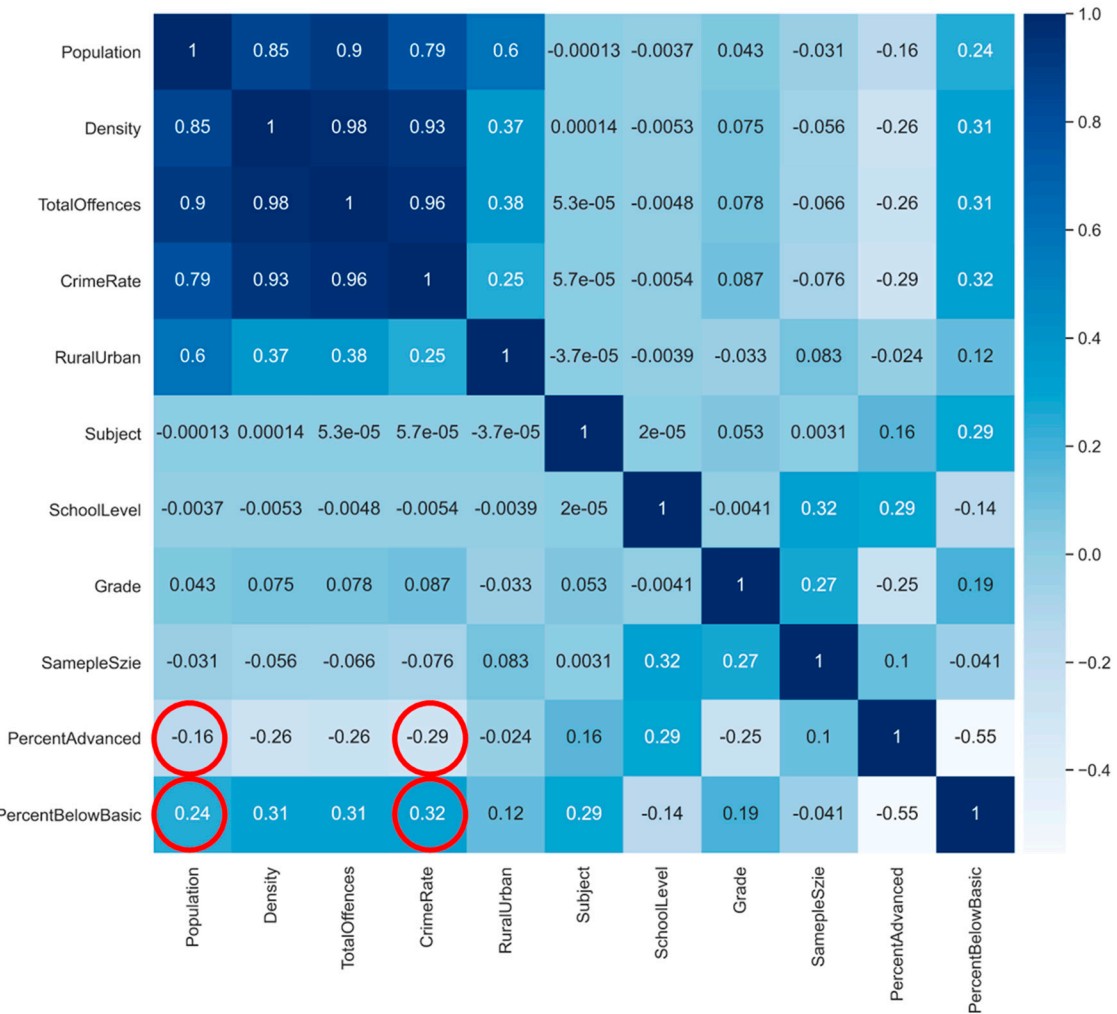

**Figure 2.** Heatmap analysis showing the relationship between "Area", "Population", "Density", "Total offences", "Crime rate", "Rural or urban", "Grade", "School level", "Sample size", "Subject", "Percentage of advanced students", and "Percentage of below-basic students". The darker the color, the stronger the positive correlation between the two data sets. The lighter the color, the stronger the negative correlation between the two data sets. The red circles visually represent the focal points of this paper's target analysis, namely the relationships between "Percentage of advanced students", "Percentage of below basic students", "Population", and "Crime rate".

Based on the heatmap, the main factors affecting academic performance (using the percentage of advanced students) were population (−0.16), density (−0.26), total offense cases (−0.26), crime rate (−0.29), rural and urban (−0.024), grade (−0.25), school level (+0.29). The lower population, lower density, lower offense cases, and lower crime rate led to a higher percentage of advanced students.

On the contrary, the higher population, higher density, higher offense cases, and higher crime rate led to a higher percentage of below-basic students. The study environment had a significant impact on the overall academic performance.

Feature importance (Figure S1) results showed that the most important factor affecting the data is the sample size (+0.458), grade (+0.126), crime rate (+0.124), subject (+0.097), and population (+0.051). The sample size was the most important factor affecting the calculation since more students had a greater impact on prediction outcomes. The crime rate and population were among the top five important factors affecting the prediction result. In the following analysis, we examined the academic impact of the crime rate and population in detail.

When we evaluated how the county population affects academic performance, the authors found the large county population (1 M~1.58 M, red box and red arrow in Figure 3) led to a lower percentage of advanced students and a higher percentage of below-basic students. The larger county population suggested a lower academic performance. On the contrary, a smaller county population helped to improve students' overall academic performance.

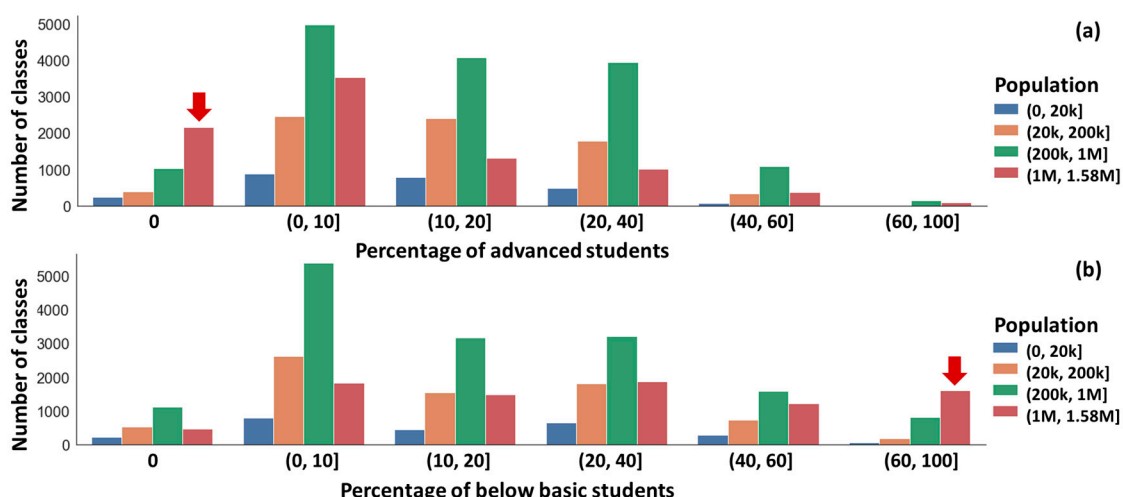

**Figure 3.** Analysis of the relationship between county population and academic performance; (**a**) percentage of advanced students; (**b**) percentage of below basic students. The red arrows indicate the highest population in the county, which are the groups specifically analyzed.

When the authors evaluated how a safe environment affected academic performance, the authors found the high crime rate (16–30 cases per 1000, red box and red arrow in Figure 4) led to a lower percentage of advanced students and a higher percentage of below-basic students. In summary, a higher crime rate led to lower academic performance. On the other side, safe environments led to higher academic performance.

The school level significantly affected academic performance (Figure 5a). In the historically underperforming schools, most of the classes only had around 2% of advanced students. In all other schools, most of the classes had around 4% of advanced students. Especially, all other schools had roughly twice as many excellent classes (more than 50% advanced students) as historically underperforming schools. Although historically underperforming schools also had some excellent classes and some good students, a higher school level significantly improved the overall academic performance. Rural schools had better overall academic performances compared to urban schools. Most rural schools had around 8% of the advanced students (red dashed line in Figure 5b), while most urban schools had only 2% of the advanced students (blue dashed line in Figure 5a).

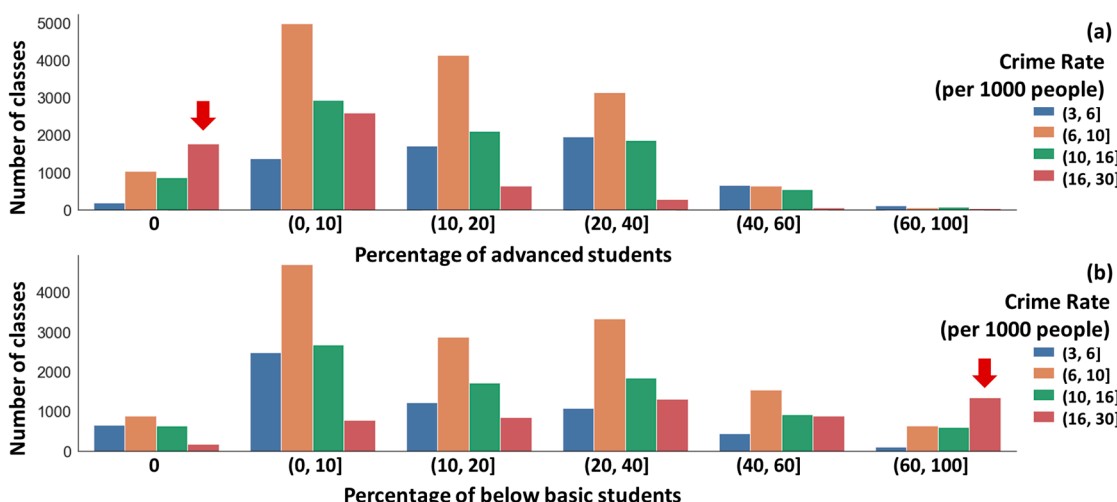

**Figure 4.** Analysis of the relationship between crime rate (per 1000 people) and academic performance; (**a**) percentage of advanced students; (**b**) percentage of below basic students. The red arrows indicate the highest crime rates, which are the groups specifically analyzed.

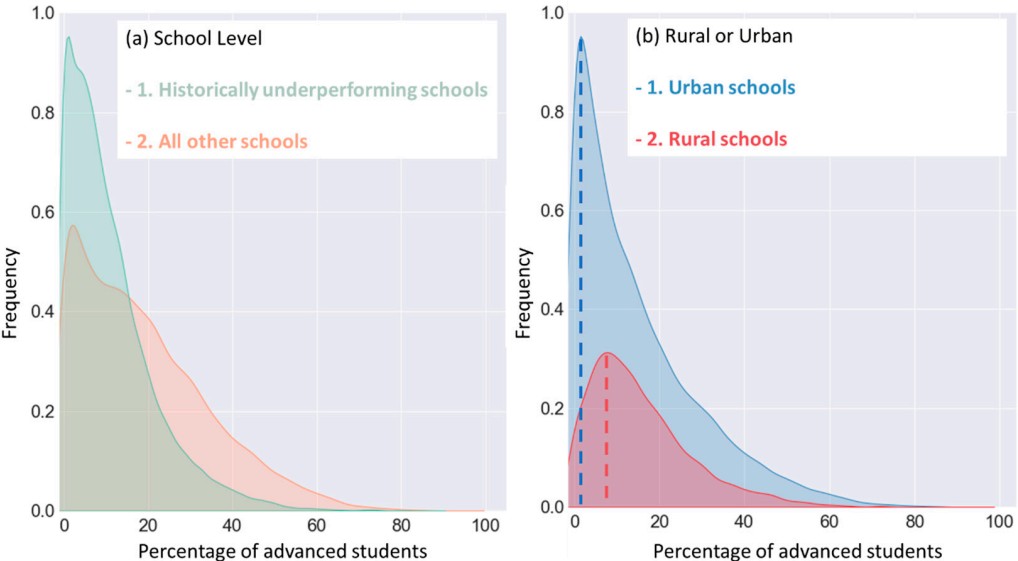

**Figure 5.** (**a**) The school level affecting academic performance (percentage of advanced students): historically underperforming schools (green line) versus all other schools (orange line); (**b**) rural or urban factors affecting academic performance: urban schools (blue line) versus rural schools (red line).

### 3.2. Academic Performance (Prediction versus Reality)

The authors evaluated five ML prediction methods, including decision tree, random forest, logistic regression, support vector machine, and neural network. Among all methods, the decision tree, random forest, logistic regression, and support vector machine achieved testing accuracy of 48%, 54%, 50%, and 51%, respectively (Table 3). The neural network achieved the highest 60% testing accuracy. As a result, this paper utilized the neural network method for the next step of the analysis.

When the authors applied the ML models, the prediction versus reality results were shown in Figure 6.

**Table 3.** Comparison of ML prediction method.

| Method | Classifier | Training Accuracy | Testing Accuracy |
| --- | --- | --- | --- |
| Decision tree | DecisionTreeClassifier | 94% | 48% |
| Random forest | RandomForestClassifier | 94% | 54% |
| Logistic regression | LogisticRegression | 48% | 50% |
| Support vector machine | SupportVectorClassifier | 59% | 51% |
| Neural network | MLPClassifier | 61% | 60% |

**Decision tree**

Worst academic performance ⟶ Best performance

| Real \ Prediciton | 0% | (0%,10%] | (10%,20%] | (20%,40%] | (40%,60%] | (60%,100%] |
| --- | --- | --- | --- | --- | --- | --- |
| 0% | **180** | 311 | 65 | 12 | 1 | 1 |
| (0%,10%] | 280 | **1554** | 499 | 134 | 20 | 3 |
| (10%,20%] | 52 | 440 | **506** | 361 | 61 | 8 |
| (20%,40%] | 11 | 142 | 387 | **769** | 340 | 87 |
| (40%,60%] | 0 | 7 | 41 | 261 | **322** | 160 |
| (60%,100%] | 0 | 1 | 18 | 107 | 415 | **573** |

**Random forest**

| Real \ Prediciton | 0% | (0%,10%] | (10%,20%] | (20%,40%] | (40%,60%] | (60%,100%] |
| --- | --- | --- | --- | --- | --- | --- |
| 0% | **145** | 367 | 45 | 12 | 0 | 1 |
| (0%,10%] | 166 | **1793** | 399 | 121 | 10 | 1 |
| (10%,20%] | 29 | 469 | **499** | 378 | 47 | 6 |
| (20%,40%] | 4 | 125 | 301 | **874** | 363 | 69 |
| (40%,60%] | 0 | 1 | 16 | 238 | **393** | 143 |
| (60%,100%] | 0 | 0 | 4 | 78 | 374 | **658** |

**Logistic regression**

| Real \ Prediciton | 0% | (0%,10%] | (10%,20%] | (20%,40%] | (40%,60%] | (60%,100%] |
| --- | --- | --- | --- | --- | --- | --- |
| 0% | **56** | 425 | 56 | 32 | 1 | 0 |
| (0%,10%] | 48 | **2063** | 214 | 145 | 20 | 0 |
| (10%,20%] | 4 | 936 | **171** | 257 | 57 | 3 |
| (20%,40%] | 1 | 553 | 211 | **828** | 137 | 6 |
| (40%,60%] | 0 | 48 | 16 | 543 | **178** | 6 |
| (60%,100%] | 0 | 4 | 1 | 146 | 207 | **756** |

**Support vector machine**

| Real \ Prediciton | 0% | (0%,10%] | (10%,20%] | (20%,40%] | (40%,60%] | (60%,100%] |
| --- | --- | --- | --- | --- | --- | --- |
| 0% | **59** | 486 | 22 | 2 | 1 | 0 |
| (0%,10%] | 36 | **2114** | 292 | 44 | 4 | 0 |
| (10%,20%] | 0 | 560 | **523** | 298 | 47 | 0 |
| (20%,40%] | 0 | 134 | 320 | **851** | 428 | 3 |
| (40%,60%] | 0 | 2 | 11 | 219 | **550** | 9 |
| (60%,100%] | 0 | 683 | 5 | 56 | 305 | **65** |

**Neural network**

| Real \ Prediciton | 0% | (0%,10%] | (10%,20%] | (20%,40%] | (40%,60%] | (60%,100%] |
| --- | --- | --- | --- | --- | --- | --- |
| 0% | **272** | 273 | 20 | 5 | 0 | 0 |
| (0%,10%] | 337 | **1877** | 208 | 63 | 5 | 0 |
| (10%,20%] | 43 | 645 | **397** | 318 | 25 | 0 |
| (20%,40%] | 4 | 163 | 263 | **978** | 307 | 21 |
| (40%,60%] | 0 | 1 | 5 | 282 | **436** | 67 |
| (60%,100%] | 0 | 0 | 0 | 21 | 196 | **897** |

**Figure 6.** Percentage of advanced students by ML models (Prediction versus reality). Bold suggests the correctly predicted. The color green indicates a relatively high numerical value, with deeper shades of green indicating even higher values. Yellow signifies intermediate numerical values. Conversely, the color red indicates relatively low numerical values, with deeper shades of red indicating even lower values.

In the decision tree, of 8129 classes, 3904 classes were correctly predicted (48% of the total data, bold in Figure 6). For the prediction that was not completely correct, 3454 classes were predicted in the neighborhood group (42% of the total data). Ten percent of the predictions were far from the real situation.

In random forest, 4362 classes (54%) were correctly predicted. For the prediction that was not completely correct, 3198 classes (39%) were predicted in the neighborhood group. Seven percent of the predictions were far from the real situation.

In logistic regression, 4052 classes (50%) were correctly predicted. For the prediction that was not completely correct, 2984 classes (37%) were predicted in the neighborhood group. Thirteen percent of the predictions were far from the real situation.

In the support vector machine, 4162 classes (51%) were correctly predicted. For the prediction that was not completely correct, 2953 classes (36%) were predicted in the neighborhood group. 13% of the predictions were far from the real situation.

The most precise prediction in this paper came from neural networks: 4857 classes (60%) were correctly predicted. For the prediction that was not completely correct, 2896 classes (36%) were predicted in the neighborhood group. Only 4% of the predictions were far from the real situation. Moreover, neural networks can predict well for all groups (both good academic performance and bad academic performance). In sum, compared with other ML models, the neural network shows good prediction stability and accuracy.

When the authors compared the real percentage of advanced students versus neural network prediction (Figure 7c versus Figure 7d), the authors found there are only minor differences between the prediction and reality, and, as a result, the authors suggested that neural network was an accurate prediction method. The authors found that the neural network prediction can find the impacts of the county population (Figure 7a) and crime rate (Figure 7b). For example, the population and crime rate are high in Philadelphia county, the neural network obtained this information and predicted a significantly lower percentage of advanced students in Philadelphia county (Figure 7d): neural network prediction matched the real situation (Figure 7c) well.

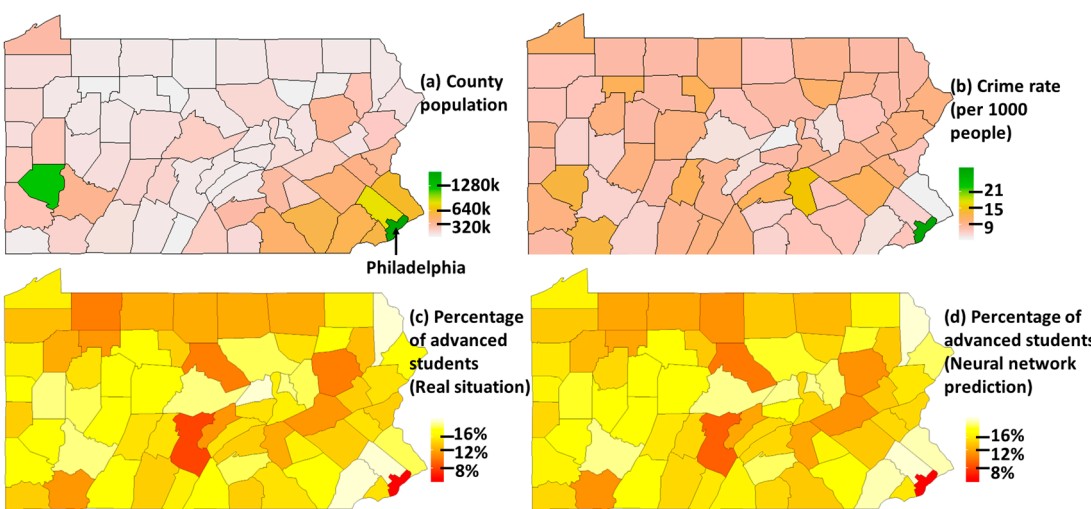

**Figure 7.** Heatmap of Pennsylvania showing (**a**) county population, (**b**) crime rate, (**c**) real percentage of advanced students, and (**d**) neural network predicted percentage of advanced students. Black arrow shows the special area of Philadelphia county.

## 4. Discussion

### 4.1. Academic Performance in Pennsylvania Schools

Some areas characterized by an inferior environment for learning (e.g., the city or county with a high population density and a considerable crime rate) could be harmful to many students, classes, and schools. The results could be explained by the less-educated people, drug abuse, poor security, gun issues, violence, and offenses. On the other hand,

well-educated people moved to a small county and created a positive learning atmosphere, better living conditions, and a supportive learning environment there, and the schools received good average academic performance. Despite this randomness of population quality within an area, the overall impact of an environment on academic performance is still recognizable. The results indicate that even with good education conditions (e.g., qualified teachers), large counties in Pennsylvania may not see a significant improvement in students' academic performance.

The reason may possibly be due to the fact that the areas with a high population density have an uneven demographic composition, varied population quality, less-educated parents, and the instabilities and uncertainties of potential offenders. Conversely, a small county with a healthy and harmonious cultural environment could contribute to raising students' academic performance in Pennsylvania schools. This could be due to the fact that these localities with well-educated people are safer and quieter and provide a healthy educational environment.

### 4.2. Advantages and Limitations of the ML Model

To predict academic performance, previous research normally used statistical models (e.g., Mplus or SPSS) to establish the relationships between such performance and SES, after which these were used in forecasting (Chang et al. 2020; Chen et al. 2021; Claver et al. 2020; Paulick et al. 2013). An example of the findings is the possibility that high income reduces parental stress, which may lead to a more stable study environment and improved academic performance (Owens 2018). The accuracy of classical methods hinges on the researchers' experience. By contrast, the current study introduced an ML model, which is simply a 'black box' that connects input (SES data) and output (academic performance) without considering relationships. The accuracy of this approach depends on data quality and quantity (Chen and Ding 2022). Through the ML method adopted in this work, academic performance in Pennsylvania's schools was successfully predicted.

The ML model is also encumbered with certain limitations, among which is its ineffectiveness in addressing the 'Black Swan' effect (Lorey et al. 2011). Most ML models generate results on the basis of data that were previously loaded into a computer program. If certain factors cannot be covered by a dataset, an ML model typically provides poor feedback. In our study, for example, for a school located in a high-density area with its surrounding environment suffering from a high crime rate, the ML model points to low academic performance. However, if this institution inputs an excellent teaching team and financial resources, it may achieve high academic performance. Moreover, there are some unaccountable factors that could not be explained by the datasets, which could cause miscalculations in an ML model.

### 4.3. Future Improvement of the ML Model

The future of ML models generally lies in two development directions: big data and novel algorithms. Considering that this is a relative feasibility analysis, it used only 33,870 records that span population, crime, and educational data (Considine and Zappalà 2002; Ginsburg and Bronstein 1993; Kurdek and Ronald 1988). In future research, if more related factors (e.g., family, economic, and transportation situations) can be considered in ML models, the authors believe such representations will generate more accurate predictions. At the same time, if data quantity can be improved (e.g., >100,000 records), more data can be used to support predictions. The availability of more data often relies on high-performance computing (Elsebakhi et al. 2015; Fox et al. 2019). If scientists in the future have access to better computers, they can also calculate more complex factors and larger amounts of data.

Algorithms are another component that can enhance ML models. In this study, five well-developed ML methods were compared: a decision tree, random forest, logistic regression, support vector machine, and neural networks. A neural network is one of the best models. The models recommended by the authors, including classification (Kotsiantis et al. 2006), KNN (Duivesteijn and Feelders 2008; Samworth 2012), linear discriminant

analysis (Izenman 2013; Xanthopoulos et al. 2013), K-means (Li et al. 2020; Likas et al. 2003), hidden Markov (Manogaran et al. 2018), and hierarchical planning (Mohr et al. 2018), can be explored by other researchers. These novel methods may also reduce computational requirements and increase predictive accuracy.

## 5. Conclusions

On the basis of big data (covering the Pennsylvania population, crime, and education data), this research demonstrated the feasibility of using ML models to predict class academic performance. To this end, the authors used an ML model that achieves fast and precise predictions: 60% of the predictions are accurate, 36% are highly close to reality, and only 4% exhibit substantial deviation from reality. This study confirmed that ML models are accurate and effective instruments. With ML models as grounding, the authors found that well-educated people in small counties that have lower crime rates could contribute to higher academic performance among Pennsylvania schools. Finally, SES exerts a significant impact on the rural–urban performance gap. The ML models are expected to provide assistance and guidance (e.g., decision making on issues that may affect performance, such as education budgets, hiring standards and practices, and teacher–student ratios) to education policymakers in the region in the future.

**Supplementary Materials:** The following supporting information can be downloaded at https://www.mdpi.com/article/10.3390/socsci12030118/s1, Figure S1: The feature importance of the factors; Figure S2: The correlation matrix of the samples; Table S1: Coding for machine learning and data analysis (Python); Table S2: Coding for feature importance (Python); Table S3: Coding for PA heatmap (R, Population, $\times 10,000$); Table S4: Coding for PA heatmap (R, Total Offenses, $\times 1000$); Table S5: Coding for PA heatmap (R, Real); Table S6. Coding for PA heatmap (R, Prediction); Table S7. Coding for PA heatmap (R, CrimeRate); Table S8. Tuning results by random search.

**Author Contributions:** Conceptualization, S.C. and Y.D.; methodology, Y.D.; software, Y.D.; validation, Y.D.; formal analysis, Y.D.; investigation, Y.D.; resources, S.C. and Y.D.; data curation, Y.D.; writing—original draft preparation, Y.D.; writing—review and editing, Y.D.; visualization, S.C.; supervision, Y.D.; project administration, Y.D. All authors have read and agreed to the published version of the manuscript.

**Funding:** This research received no external funding.

**Acknowledgments:** The authors thank the knowledge and computation support from the School of Geography and the Environment, University of Oxford, United Kingdom.

**Conflicts of Interest:** The authors declare no conflict of interest.

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
