# Peer review of "A Machine Learning Approach to Predicting Academic Performance in Pennsylvania’s Schools"

_socsci, doi:10.3390/socsci12030118_

Round 1
Reviewer 1 Report
The presentation flow, empirical results are consistent with the literature in the related research field. The machine learning algrithms used are basically the traditional ones. I did not see any new innovation on the algorithmic design. However, it would be acceptable for a non-machine journal. The results obtained are based on historical data. It will be interesting to look into the progressive temporal data as well. There are few studies that examine the prediction of student academic performance published in the Journal of Educational Computing Research.
Author Response
Reviewer 1:
The presentation flow, empirical results are consistent with the literature in the related research field. The machine learning algrithms used are basically the traditional ones. I did not see any new innovation on the algorithmic design. However, it would be acceptable for a non-machine journal. The results obtained are based on historical data. It will be interesting to look into the progressive temporal data as well. There are few studies that examine the prediction of student academic performance published in the Journal of Educational Computing Research.
Authors’ response:
We appreciate the reviewer's feedback on our study. While we acknowledge that our study does not introduce any novel algorithmic design, we believe that our study's contribution lies in the application of traditional machine learning algorithms to the academic performance prediction task. By demonstrating the feasibility and effectiveness of using these algorithms in this context, our study highlights the potential of machine learning as a valuable tool for improving education policy design.
Our research has successfully established a machine learning model for predicting academic performance and demonstrated its moderate level of accuracy. This achievement is particularly noteworthy because it opens up a new approach to academic performance prediction, which has traditionally relied on classical machine learning programs. We hope that this study will inspire other researchers to explore the potential of machine learning in education and contribute to the development of more effective tools for education policy design.
Authors’ related revision:
(Line 5-Line 16) Abstract: Academic performance prediction is an indispensable task for policymakers. Academic performance is frequently examined using classical statistical software, which can be used to detect logical connections between socioeconomic status and academic performance. These connections, whose accuracy depends on researchers’ experience, determine prediction accuracy. To eliminate the effects of logical relationships on such accuracy, this research used ‘black box’ machine learning models extended with education and socioeconomic data on Pennsylvania to predict academic performance in the state. The decision tree, random forest, logistic regression, support vector machine and neural network achieved testing accuracies of 48%, 54%, 50%, 51% and 60%, respectively. The neural network model can be used by policymakers to accurately forecast academic performance, which in turn can aid in the formulation of various policies, such as those on funding and teacher selection. Finally, this study demonstrated the feasibility of machine learning as an auxiliary educational decision-making tool for use in the future.
Reviewer 2 Report
This paper studies the task of academic performance prediction. Since logical relationships are often considered such as socioeconomic status etc., which tends to bring in bias, this paper uses a neural network model.
The case study is based in Pennsylvania since Pennsylvania is a representative state including big cities like Pittsburgh and Philadelphia and rural areas, consisting of wide range of educational resources.
The NN is designed to consist of 100 hidden layers with 100 nodes each. The maximum iterations was set at 50. The results show promising prediction accuracies when compared with the actual numbers.
It would be nice to see a wider comparison with respect to covering larger areas and case studies as well as other machine learning models involving more feature input variables modeling the exogenous variables as well.
Author Response
Reviewer 2:
This paper studies the task of academic performance prediction. Since logical relationships are often considered such as socioeconomic status etc., which tends to bring in bias, this paper uses a neural network model.
The case study is based in Pennsylvania since Pennsylvania is a representative state including big cities like Pittsburgh and Philadelphia and rural areas, consisting of wide range of educational resources.
The NN is designed to consist of 100 hidden layers with 100 nodes each. The maximum iterations was set at 50. The results show promising prediction accuracies when compared with the actual numbers.
It would be nice to see a wider comparison with respect to covering larger areas and case studies as well as other machine learning models involving more feature input variables modeling the exogenous variables as well.
Authors’ response:
Thank you for your review. We would like to clarify that this paper is a feasibility study that explores the use of machine learning (ML) methods for academic performance prediction. The study design is based on computational approaches, which are known for their efficiency and ability to handle large amounts of data. However, due to computational limitations, we focused our analysis on a specific case study in Pennsylvania.
We acknowledge that the use of more powerful computing resources in the future could potentially expand the scope of our analysis to cover larger areas and case studies. Additionally, the inclusion of more input variables in the ML model could improve the accuracy of our predictions. We appreciate your suggestion for further research in this area, and we hope that our study will contribute to the development of more accurate and efficient academic performance prediction methods in the future.
Authors’ related changes:
(Line 5-Line 16) Abstract: Academic performance prediction is an indispensable task for policymakers. Academic performance is frequently examined using classical statistical software, which can be used to detect logical connections between socioeconomic status and academic performance. These connections, whose accuracy depends on researchers’ experience, determine prediction accuracy. To eliminate the effects of logical relationships on such accuracy, this research used ‘black box’ machine learning models extended with education and socioeconomic data on Pennsylvania to predict academic performance in the state. The decision tree, random forest, logistic regression, support vector machine and neural network achieved testing accuracies of 48%, 54%, 50%, 51% and 60%, respectively. The neural network model can be used by policymakers to accurately forecast academic performance, which in turn can aid in the formulation of various policies, such as those on funding and teacher selection. Finally, this study demonstrated the feasibility of machine learning as an auxiliary educational decision-making tool for use in the future.
(Line 274-289) The future of ML models generally lies in two development directions: big data and novel algorithms. Considering that this is a relative feasibility analysis, it used only 33,870 records that span population, crime and educational data [43-45]. In future research, if more related factors (e.g. family, economic and transportation situations) can be considered in ML models, the authors believe such representations will generate more accurate predictions. At the same time, if data quantity can be improved (e.g. >100,000 records), more data can be used to support predictions. The availability of more data often relies on high-performance computing [46,47]. If scientists in the future have access to better computers, they can also calculate more complex factors and larger amounts of data.
Algorithms are another component that can enhance ML models. In this study, five well-developed ML methods were compared: a decision tree, random forest, logistic regression, support vector machine and neural network. A neural network is one of the best models. The models recommended by the authors, including classification [48], KNN [49,50], linear discriminant analysis [51,52], K-means [53,54], hidden Markov [55] and hierarchical planning [56], can be explored by other researchers. These novel methods may also reduce computational requirements and increase predictive accuracy.
Reviewer 3 Report
The title as well as the introduction raised expectations about your manuscript and research. The topic you are addressing would be a relevant addition to existing literature. Thank you for this valuable contribution. I will structure my feedback in (a) general remarks (these comments cover feedback applicable in the entire manuscript), and (b) specific remarks (feedback on sentence and/or word level). The specific remarks can include a quote from your original manuscript to refer to a specific section. The specific remarks will refer to page (emphasis added in boldface; e.g., 1.15/16) and row(s; e.g., 11.15/16).
General remarks:
The overall manuscript is neat and written concisely—with relevant information for existing literature. One aspect that you can focus on is consistency in terminology and concise writing.
Specific remarks:
p.1.5/6 “programs, through which…” à Sentence construction. It reads a bit odd. Moreover, if you call those relationships “relationships” stick with that (do not call them connections in the sentence after). Keep your essential terminology consistent.
p.1.9. “as well as” = extended with. The construction as it is now implies that you also work with educational models.
p.1.10 forecast = predict (essential terms).
p.1.33 “Kryst et al.” = redundant.
p.1 You are using academic achievement and academic performance interchangeably.
p.2.47–62 Explain what ML does and insert the references between square brackets. The examples in another context do not work here (it is irrelevant and confusing).
p.2.59 Do all the aforementioned have all criteria?
p.3.78/79 SES and crime rates are variables on a much larger scale than, for example, inter- and intrapersonal factors. To provide a complete overview, this has to be mentioned. That your scope will be on macro-level variables can be announced.
p.Table2 Delete the vertical lines (see your Table 1). I would present this information differently. This is hard to read.
p.figure2 This can be presented larger/bigger. In a similar vein, Figure 4 can be presented larger/bigger to make it easier to read.
p.figures Some of these can be moved to the Appendices.
p.references There are inconsistencies in displaying the references. Compare reference 13 with references 3.
Author Response
Reviewer 3:
The title as well as the introduction raised expectations about your manuscript and research. The topic you are addressing would be a relevant addition to existing literature. Thank you for this valuable contribution. I will structure my feedback in (a) general remarks (these comments cover feedback applicable in the entire manuscript), and (b) specific remarks (feedback on sentence and/or word level). The specific remarks can include a quote from your original manuscript to refer to a specific section. The specific remarks will refer to page (emphasis added in boldface; e.g., 1.15/16) and row(s; e.g., 11.15/16).
General remarks:
The overall manuscript is neat and written concisely—with relevant information for existing literature. One aspect that you can focus on is consistency in terminology and concise writing.
Authors’ response: the authors used ML for “machine learning”, and SES for “socioeconomic status”, STEM for “science, technology, engineering, and mathematics”. Other places are used the full spellings.
Specific remarks:
p.1.5/6 “programs, through which…” à Sentence construction. It reads a bit odd. Moreover, if you call those relationships “relationships” stick with that (do not call them connections in the sentence after). Keep your essential terminology consistent.
Authors’ response: Thank you for your suggestions. The authors have rewritten the sentence to enhance its cohesion.
Authors’ change (Line 5): Academic performance prediction is an indispensable task for policymakers. Academic performance is frequently examined using classical statistical software, which can be used to detect logical connections between socioeconomic status and academic performance. These connections, whose accuracy depends on researchers’ experience, determine ...
p.1.9. “as well as” = extended with. The construction as it is now implies that you also work with educational models.
Authors’ response: Thank you for your suggestions. The authors made the change based on the suggestion.
Authors’ change (Line 9): as well as -> extended with
p.1.10 forecast = predict (essential terms).
Authors’ response: Thank you for your suggestions. The authors made the change based on the suggestion.
Authors’ change (Line 10): forecast -> predict
p.1.33 “Kryst et al.” = redundant.
Authors’ response: Thank you for your suggestions. The authors made the change based on the suggestion.
Authors’ change (Line 33): In eastern Europe, for example, Kryst et al. found support for the argument that education from rural schools and low SES can lead to a disadvantage in student performance [5]. -> In eastern Europe, the education from rural schools and low SES can lead to a disadvantage in student performance [5].
p.1 You are using academic achievement and academic performance interchangeably.
Authors’ response: Following the reviewers’ suggestion, the authors used performance only.
Authors’ change: achievement -> performance
p.2.47–62 Explain what ML does and insert the references between square brackets. The examples in another context do not work here (it is irrelevant and confusing).
p.2.59 Do all the aforementioned have all criteria?
Authors’ response: Following the reviewers' suggestion, the authors have deleted the related examples that were causing confusion.
Authors’ change: (Line 47-62) deletion of examples.
p.3.78/79 SES and crime rates are variables on a much larger scale than, for example, inter- and intrapersonal factors. To provide a complete overview, this has to be mentioned. That your scope will be on macro-level variables can be announced.
Authors' response: The authors revised the manuscript based on the reviewers' suggestions.
Authors' change: (Line 78) This research focused on the effects of socioeconomic status (SES) and crime rates on school performance. SES and crime rates are variables on a much larger scale than, for example, interpersonal and intrapersonal factors. We chose Pennsylvania as the basis for our study.
p.Table2 Delete the vertical lines (see your Table 1). I would present this information differently. This is hard to read.
p.figure2 This can be presented larger/bigger. In a similar vein, Figure 4 can be presented larger/bigger to make it easier to read.
p.figures Some of these can be moved to the Appendices.
p.references There are inconsistencies in displaying the references. Compare reference 13 with references 3.
Authors' response: The authors have revised the manuscript in response to the reviewers' suggestions.
Authors' change: The authors have deleted the vertical lines of Table 2 and increased the size and resolution of figures to make them easier to read. They have also edited the related reference.
